# LiMMCov: An interactive research tool for efficiently selecting covariance structures in linear mixed models using insights from time series analysis

**Perseverence Savieri**[1,2]*, **Lara Stas**[1,2], **Kurt Barbé**[1,2]

**1** Biostatistics and Medical Informatics Research Group (BISI), Vrije Universiteit Brussel (VUB), Brussels, Belgium, **2** Core Facility - Support for Quantitative and Qualitative Research (SQUARE), Vrije Universiteit Brussel (VUB), Brussels, Belgium

* Perseverence.Savieri@vub.ac.be

## Abstract

The correct specification of covariance structures in linear mixed models (LMMs) is critical for accurate longitudinal data analysis. These data, characterised by repeated measurements on subjects over time, demand careful handling of inherent correlations to avoid biased estimates and invalid inferences. Incorrect covariance structure specification can lead to inflated type I error rates, reduced statistical power, and inefficient estimation, ultimately compromising the reliability of statistical inferences. Traditional methods for selecting appropriate covariance structures, such as AIC and BIC, often fall short, particularly as model complexity increases or sample sizes decrease. Studies have shown that these criteria can misidentify the correct structure, resulting in suboptimal parameter estimates and poor assessment of standard errors for fixed effects. Additionally, relying on trial-and-error comparisons in LMMs can lead to overfitting and arbitrary decisions, further undermining the robustness of model selection and inference. To address this challenge, we introduce LiMMCov, an interactive app that uniquely integrates time-series concepts into the process of covariance structure selection. Unlike existing tools, LiMMCov allows researchers to explore and model complex structures using autoregressive models, a novel feature that enhances the accuracy of model specification. The app provides interactive visualisations of residuals, offering insights into underlying patterns that traditional methods may overlook. LiMMCov facilitates a systematic approach to covariance structure selection with a user-friendly interface and integrated theoretical guidance. This paper details the development and features of LiMMCov, demonstrates its application with an example dataset, and discusses its potential impact on research. The app is freely accessible at https://zq9mvv-vub0square.shinyapps.io/LiMMCov-research-tool/.

**Data availability statement:** All relevant data are within the manuscript and its Supporting information files.

**Funding:** The author(s) received no specific funding for this work.

**Competing interests:** The authors have declared that no competing interests exist.

## 1. Introduction

Linear mixed models (LMMs) have been extensively studied and utilised to analyse longitudinal data in various fields, including medical research and social sciences [1,2]. Longitudinal data is collected on subjects or sampling units repeatedly over time [3,4]. Since these repeated measurements are positively correlated, ignoring this correlation in the analysis can lead to underestimated standard errors and inflated type I error rates, as well as potentially incorrect type II error rates [5]. LMMs address the dependency in the data by incorporating both fixed effects (consistent across individuals) and random effects (which vary among individuals or clusters) [3]. Verbeke and Molenberghs [2] represent the mixed model for observed outcome data ($y_i$) as:

$$y_i = X_i\beta + Z_ib_i + \varepsilon_i \tag{1}$$

$$b_i \sim N(0, D), \quad \varepsilon_i \sim N(0, \Sigma_i)$$

$X$ is a design matrix for the fixed effects $\beta$, $Z$ is the design matrix for random effects $b_i$, and $\varepsilon$ is the residual error vector. In this model, $b_i$ and $\varepsilon_i$ are assumed independent, D represents the random effects covariance matrix, and $\Sigma_i$ represents the covariance matrix of the residual errors. The model captures correlations using two approaches: 1) Hierarchical formulation (i.e., the conditional or subject-specific model) and 2) Marginal formulation (i.e., the population-average or marginal model). In the first approach, given the random effects, the measurements of each subject are independent (conditional independence assumption). This means that the more random effects we include, the more flexibly we capture the correlations [2,6]. In the second approach, the measurements of each subject are correlated, and these correlations between repeated measurements are modelled by the marginal covariance matrix $V_i = Z_iDZ_i' + \Sigma_i$ [2,3,7]. A critical component of this marginal formulation is the correct specification of the covariance matrix [8]. This matrix ($V_i$) can be explicitly modelled to reflect any remaining correlation among residuals after specifying time-varying fixed effects, or it can be left unstructured. Common choices for modelling $V_i$ include the autoregressive model of order 1, AR(1), which assumes that correlations decay geometrically over time (or repeated measures) and compound symmetry (CS), which assumes a uniform correlation among all pairs of observations within the same subject [2,4,9]. In contrast, leaving $V_i$ unstructured means that the data specify all variances and covariances without constraining structures and ignoring residual correlation when it exists can lead to biased estimates and misinformed inference, highlighting the importance of proper evaluation of $V_i$ [1,2]. In this study, we focus on the marginal formulation where the covariance structure is modelled by $V_i$.

Despite the critical importance of an appropriate structure for $V_i$, many researchers remain unaware of its impact and often rely on default settings, making the selection process challenging. As Molenberghs & Verbeke [2] emphasise, "while many residual covariance structures are available, there is a lack of general simple techniques for model comparisons". The current body of literature has explored various

methods for covariance structure selection, including model comparison techniques and diagnostic tools [8]. Researchers typically use the likelihood ratio test (LRT) to compare a pair of nested models, i.e., "full" and "reduced" models, where the "reduced" model's covariance structure is a special case of the "full" model, e.g., when comparing the unstructured with the CS [5,10]. However, LRTs are limited by their requirement for nested models and their sensitivity to sample size, leading to challenges in model comparison and a higher risk of overfitting or failing to detect true differences [3]. For non-nested models, information criteria, such as Akaike's Information Criterion (AIC; [11]), AIC Corrected Criterion (AICc; [12]), Consistent AIC (CAIC; [13]) and the Bayesian Information Criterion (BIC; [14]) are employed. The effectiveness of these information criteria in selecting the correct residual covariance structure has been studied by Ferron et al. [15], Gómez et al. [16], and Keselman et al. [17]. Unfortunately, these criteria do not always succeed in selecting the proper structure. Ferron et al. [15] found that AIC selected the correct covariance structure about 79% of the time, while BIC did so 66% of the time. In contrast, Keselman et al. [17] reported that AIC and BIC were successful only 47% and 35% of the time. However, while the AIC is useful for comparing models, it offers limited insight into why one model outperforms another, effectively making the selection process a black box for identifying the model structure. The selection accuracy decreases as the covariance structure becomes complex and the sample size decreases. Although Vallejo et al. [8] recommend that researchers may benefit from using criteria designed for small samples, such as AICc, the inferential properties of such criteria remain underexplored. Other studies have shown that information criteria like CAIC and BIC can be competitive in identifying the correct covariance structure when paired with simulation-based approaches [18,19]. These findings are context-dependent and limited to specific simulation settings. The conclusion that these criteria perform well under certain conditions does not address the broader challenges faced in real-world applications, where model complexity and data variability often undermine their effectiveness. Thus, one must consider the possible effects of misspecification of the covariance structure on the statistical properties of the inferences.

Misspecifying the marginal covariance structure can have significant consequences when analysing repeated measurement data [8,15,20–22]. Incorrect selection leads to inefficient estimation and poor assessment of standard errors for fixed effects [2,20,23–25], leading to biased statistical inferences, inflated type I error rates, and reduced statistical power [8,20,21]. Although the fixed effect estimates often remain unbiased [15], selecting an inadequate covariance structure can compromise the control over type I error rates associated with hypothesis tests for fixed effects, such as group differences or time trends [16,26–28]. This issue becomes pronounced with smaller sample sizes, fewer repeated measures, and more complex covariance structures [8]. Additionally, properly modelling the marginal covariance structure captures the remaining temporal dependencies and ensures more accurate estimates of the variability in the outcome over time after accounting for fixed effects [29].

There is a clear need for an accessible and detailed approach to selecting covariance structures in LMMs for longitudinal data to avoid the consequences of misspecifications. Although commercial software procedures such as SAS Proc Mixed and SPSS Mixed offer a menu of covariance structures, these platforms are not accessible to everyone. Traditional methods described by Littell et al. [9] provide limited guidance and visualisation options, often leaving users to rely on trial and error or suboptimal default settings. Despite research on graphical procedures such as the draftsman's display and parallel axis plots [30], there is still a significant gap in understanding and accurately identifying complex covariance structures in longitudinal data, mainly when these structures are not easily discernible through traditional visualisation methods. Moreover, these methods often require advanced statistical knowledge and are not always user-friendly. Existing research tools for LMMs do not include features to explicitly model the marginal covariance structure [31]. Therefore, researchers require a free tool that provides robust statistical methods and offers an intuitive interface, theoretical guidance, and interactive visualisations to aid in selecting the correct marginal covariance structures.

A key insight of this study is leveraging time-series analysis concepts to address the challenge of modelling correlation structures in longitudinal data. Specifically, when random effects or a marginal covariance matrix are not included, the correlation remains in the residuals [32]. This observation provides a foundation for visualising and understanding residual

correlation patterns, enabling researchers to select appropriate residual covariance structures more effectively. Another novel feature of this study is the application of Yule-Walker equations to construct correlation matrices [33], which were employed to simulate datasets and validate the proposed method. These insights form the basis of our interactive tool, **LiMMCov**, developed to make these advanced concepts accessible to a broader audience.

LiMMCov is an interactive research tool built using the R Shiny package [34] to intuitively guide researchers in selecting appropriate covariance structures for longitudinal data analysis. It integrates interactive visualisations of residuals and applies time-series concepts, such as autoregressive models, to model complex correlation structures. The app's user-friendly design, point-and-click interface, and detailed theoretical text allow researchers to load their own or example datasets, visualise data properties, and apply preprocessing techniques such as variable transformations and data type changes. By bridging the gap between statistical theory and practical application, LiMMCov has the potential to enhance the analytical capabilities of researchers, ensuring more accurate and efficient longitudinal data analyses.

In the Methods section, we summarise the time-series concepts that motivated our work, including all statistical details. We describe the design and features of LiMMCov, and the correlation structures included. Readers primarily interested in the practical applications can skip to section 2.4 and read the Results section, where we demonstrate the capabilities of LiMMCov through a step-by-step guide using an example dataset from published research. The Discussion section then highlights the tool's strengths and limitations, followed by concluding remarks.

## 2. Methods

### 2.1 Process for evaluating the proposed method

**2.1.1 Deriving correlation matrices using Yule-Walker equations.** Time series analysis is used to analyse data points collected or observed at successive points in time [35]. These analyses aim to understand underlying patterns, forecast future values, or model the dependency structure within the data. A key concept in time series analysis is the autocorrelation function, which quantifies the degree of correlation between observations at different time lags [32]. Within this framework, autoregressive (AR) models are frequently used to capture dependencies between current and past observations in a time series.

The AR(p) model is given by the equation:

$$X_t = \phi_1 X_{t-1} + \phi_2 X_{t-2} + \ldots + \phi_p X_{t-p} + \varepsilon_t \tag{2}$$

where $X_t$ represents the time series at time $t$, $\phi_i$ are the autoregressive coefficients, p is the order of the AR process, and $\varepsilon_t$ is a white noise error term with variance $\sigma^2$.

As shown in model (2), AR models assume that each observation can be expressed as a linear combination of its preceding values, along with a random error term [32,35].

The Yule-Walker equations are fundamental in the estimation of AR model parameters (i.e., autoregressive coefficients, $\phi_i$). These equations establish a relationship between the autoregressive coefficients and the autocorrelation function of the time series, allowing the AR parameters to be derived directly from the data [33].

In this study, we utilised the Yule-Walker equations to estimate the autoregressive coefficients required to construct correlation matrices for AR processes.

The Yule-Walker equations for an AR(p) process are given by:

$$\gamma_k = \sum_{i=1}^{p} \phi_i \gamma_{k-i} + \sigma^2 \delta_{k0}, \qquad k = 0, 1, 2, \ldots, p \tag{3}$$

where $k=0,\ldots,p$ yielding $p+1$ equations [36]. Each term in the equation is defined as follows:

- $\gamma_k$: The autocovariance at lag $k$ which quantifies the similarity between the time series values separated by $k$ time steps. It reflects the degree of dependency between observations at different points in time.

- $\phi_i$ :The autoregressive (AR) coefficients, which represent the contribution of the $i$-th lagged value to the current value in the time series. These are the parameters to be estimated in the AR model.

- $\sigma^2$ :The variance of the input noise process. It captures the variability in the time series that is not explained by its past values.

- $\delta_{k0}$ is the Kronecker delta, which is 1 for $k = 0$ and 0 otherwise. It ensures that the noise variance contributes only to the autocovariance at lag $k = 0$ (i.e., the variance of the series).

- $p$: The order of the AR process, indicating how many past time steps are considered when modelling the current value.

Since the autocorrelation function (ACF) $\rho_k$ is defined as the autocovariance at lag $k$ normalized by the variance $\gamma_0$:

$$\rho_k = \frac{\gamma_k}{\gamma_0} \tag{4}$$

the Yule-Walker equations can be rewritten in terms of the autocorrelation function to give model (5):

$$\rho_k = \sum_{i=1}^{p} \phi_i \rho_{k-i}, \qquad k = 1, 2, \ldots, p \tag{5}$$

Matrix form *of* Yule-Walker equations

To enhance the theoretical clarity of parameter estimation, the Yule-Walker equations for an AR(p) process can be expressed in matrix form as:

$$\begin{bmatrix} \rho_0 & \rho_1 & \cdots & \rho_{p-1} \\ \rho_1 & \rho_0 & & \rho_{p-2} \\ \vdots & & \ddots & \vdots \\ \rho_{p-1} & \rho_{p-2} & \cdots & \rho_0 \end{bmatrix} \begin{bmatrix} \phi_1 \\ \phi_2 \\ \vdots \\ \phi_p \end{bmatrix} = \begin{bmatrix} \gamma_1 \\ \gamma_2 \\ \vdots \\ \gamma_p \end{bmatrix}$$

or concisely

$$\boldsymbol{R}\Phi = \Gamma \tag{6}$$

where $\boldsymbol{R}$ is the $p$ x $p$ autocorrelation matrix, $\Phi$ is the $p$ x $1$ vector of AR coefficients and $\Gamma$ is the $p$ x $1$ vector of autocorrelations. Yule-Walker equations for AR(1) and AR(2) processes are provided in the S1 File.

To derive the correlation matrices $\boldsymbol{R}$ in our study, we first estimated autoregressive (AR) parameters from weakly stationary time series via the Yule-Walker equations, confirming that all characteristic roots lay outside the unit circle. Once these AR parameters were estimated, we constructed correlation matrices for different AR processes with specific dimensions based on the number of repeated measurements. For AR(2) processes, we utilised a modified version of the *mat.ar2()* function from the **dae** package [37], to generate the corresponding correlation matrix. The modified version includes additional checks for AR(2) model causality and provides a more precise computation for the third lag, ensuring greater model stability and accuracy. The complete R code used to implement this whole process can be found in the supplementary information S2.

## 2.2 Numerical validation

We simulated longitudinal data incorporating specified correlation structures (i.e., CS, AR1 and AR2) to create a controlled yet realistic environment for evaluation. The simulated datasets were based on previously published longitudinal studies on HIV-positive adults [38–40] to reflect standard demographic and clinical variables encountered in real-world longitudinal studies. This ensured that the correct sample covariance structure was known, allowing us to systematically evaluate

the accuracy of different modelling approaches. These datasets, with realistic longitudinal patterns, provided a robust testing environment for our methods and are included in the app as examples. Details of the R code used to generate these datasets can be found in S2 File.

## 2.3 Fitting linear models and visualising residuals

### 2.3.1 Importance of residual analysis.
One of the key insights from time series analysis is that when a model does not account for the correlation inherent in structured data, this correlation manifests in the residuals [33]. Residual patterns can reveal temporal dependencies, clustering, or other correlation structures that a given model may have missed. This makes examining residuals a powerful diagnostic tool to ensure the adequacy of the model's assumptions and to guide the choice of appropriate correlation structures.

### 2.3.2 Initial model fitting.
In this study, we initially fitted linear models to the simulated data using the generalised least squares method without specifying any random effects or correlation structures. This step was deliberate: by focusing solely on the fixed effects and neglecting the clustering or correlation in the data, we could assess the linear models to reveal the underlying structure. Residuals from these fixed effects models were extracted, and their covariance and correlation matrices were computed. In these steps, we aimed to identify the structure depicted in the correlation matrices.

### 2.3.3 Visualising residual correlation.
To visualise the temporal correlation structure, we plotted residuals as means of the off diagonals in the correlation matrix against the lag. This approach enabled us to uncover the remaining correlation patterns that the initial linear model did not account for. Ideally, residuals in a well-specified fixed effects model should be independent and randomly distributed. However, patterns or trends in the correlation matrices of the residuals indicated that the model had failed to capture some aspects of the underlying structure (see plots in Fig 1).

Visualising these residuals against the lag provided valuable insights into the nature of the correlation, for example, identifying whether the pattern was autoregressive or compound symmetric. By incorporating these insights, subsequent models could more accurately reflect the true correlation structure, avoiding reliance on purely theoretical assumptions or a trial-and-error approach. This step ensured that the analysis remained robust and grounded in empirical evidence.

### 2.3.4 Using PACF to detect autoregression order.
In addition to residual plots, we used the Partial Autocorrelation Function (PACF) plots to detect autocorrelation and determine the appropriate order of the autoregressive (AR) correlation structures (Fig 2). The PACF isolates the direct relationship between residuals at specific lags, filtering out the influence of shorter lags.

Residual plots provide initial insights into autoregressive patterns, such as a decaying trend (often indicative of an odd order) or oscillations (suggesting an even order). However, they do not explicitly determine the true AR order. For instance, a decaying trend in residuals might result from the combined effects of three AR(1) components, effectively forming an AR(3) structure. This ambiguity highlights the need for PACF plots, which provide a more precise indication of the correct AR order by focusing exclusively on direct correlations at each lag.

In our context, PACF plots were particularly valuable compared to the Autocorrelation Function (ACF), which measures overall correlations at various lags. The ACF can mislead by suggesting longer lag correlations due to the cumulative effects of intermediate lags. By filtering out these confounding effects, the PACF simplified the process of identifying the correct AR order (i.e., AR(1), AR(2), or higher), ensuring that our models more accurately reflected the underlying correlation structure [33, 36].

## 2.4 User interface overview

LiMMCov was developed in R [41] and implemented as a web application using the R Shiny package [34]. It is available as an open-source application at https://zq9mvv-vub0square.shinyapps.io/LiMMCov-research-tool/. A complete list of the R packages that were utilised can be found in the source code which is available on GitHub at https://github.com/vub-square/LiMMCov-Shiny-app.

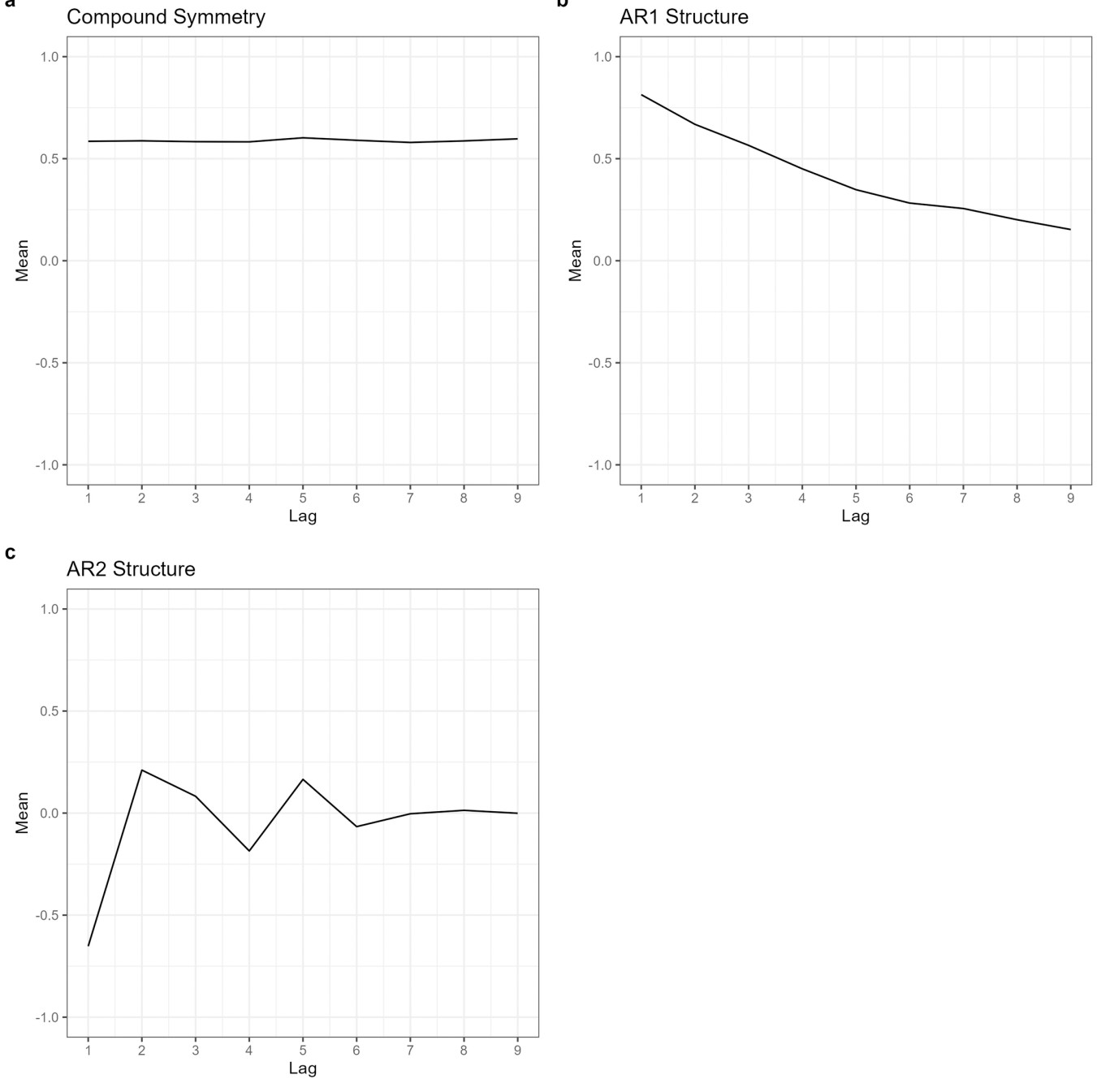

**Fig 1. Residual plots.** Residuals illustrating correlation structures from linear models: (a) compound symmetry, (b) autoregressive order 1 (AR1) and (c) autoregressive order 2 (AR2).

The *Home Tab* introduces the app and outlines its purpose and functionality ([Fig 3]). The app contains several tabs dedicated to a specific part of the analysis workflow under the *Analysis* navigation menu.

The *Dataset Tab* allows users to select an example dataset for analysis. In addition to the simulated datasets, we included a dataset named "COVID data," drawn from a published prospective observational study of patients with severe

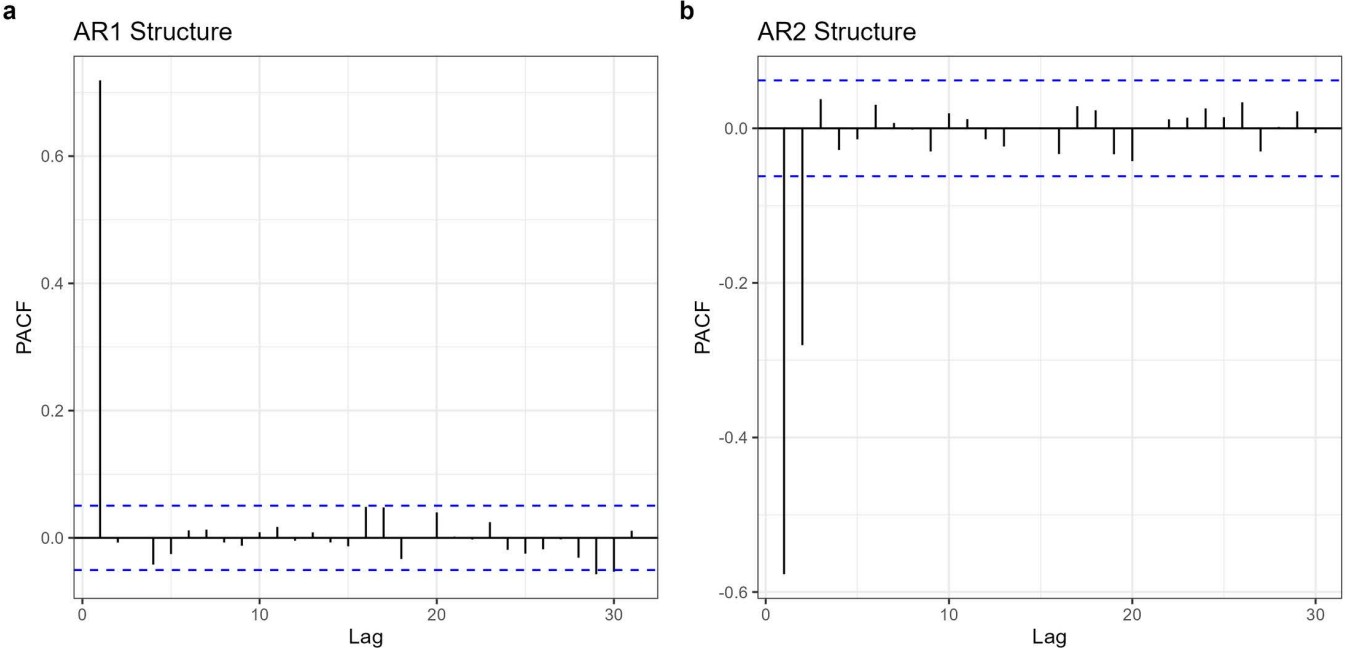

**Fig 2. PACF plots.** An illustration of autocorrelation structures from linear models: (a) autoregressive order 1 and (b) autoregressive order 2.

acute respiratory syndrome coronavirus 2 (SARS-CoV-2) infection admitted to the intensive care unit (ICU) at a tertiary hospital in Cape Town, South Africa [42]. The dataset includes 413 patients admitted between 27 March and 4 November 2020. Clinical and laboratory data were captured daily and stored electronically, including demographic and lifestyle characteristics, clinical disease characteristics, pre-existing comorbidities, arterial blood gases, and routinely collected laboratory data. The primary outcomes analysed in the study were mortality rates and length of stay in the ICU, with time to death or discharge as secondary outcomes. Users can also upload their own datasets by specifying the data format (e.g., CSV, Excel) and file location. Once data has been loaded, there are options to transform variables or change data types under *"Data manipulation and preprocessing"*. The "Data Summary" includes basic descriptive statistics and visualisations to understand the data distribution.

The *General Linear Model Tab* enables users to fit a general linear model (GLM) to the dataset. Users can specify the outcome variable and the mean structure appropriate for their data. The mean structure will include the fixed effects of the model. A summary of the fitted GLM model is then provided on the main panel.

The *Covariance Analysis Tab* then provides tools for analysing the correlation structure of the residuals from the GLM. Users can visualise 1) residual plots and 2) partial autocorrelation function (PACF) plots which can easily be downloaded for further exploration. This tab also includes the interpretation of ideal residual and PACF plots for common correlation structures. In a compound symmetry (CS) structure, all residuals are equally correlated, irrespective of lag, resulting in a horizontal line at the value rho ($\rho$) in the residual plot. The PACF shows a single significant spike at lag 1, indicating a constant correlation across all time points without a decay pattern. In AR(1), the correlation between observations decreases exponentially with lag distance. Therefore, residuals show a declining line in the residual plot as correlation decreases exponentially with increasing lag. The PACF displays a significant spike at lag 1, with subsequent lags showing values close to zero, reflecting the single-level decay of correlation over time. In AR(2), two autoregressive (AR) parameters influence the correlation pattern, reflecting the interaction between two lag terms. Residuals oscillate around zero, reflecting the interplay between lag terms without systematic bias. The PACF typically shows significant spikes at lags 1 and 2,

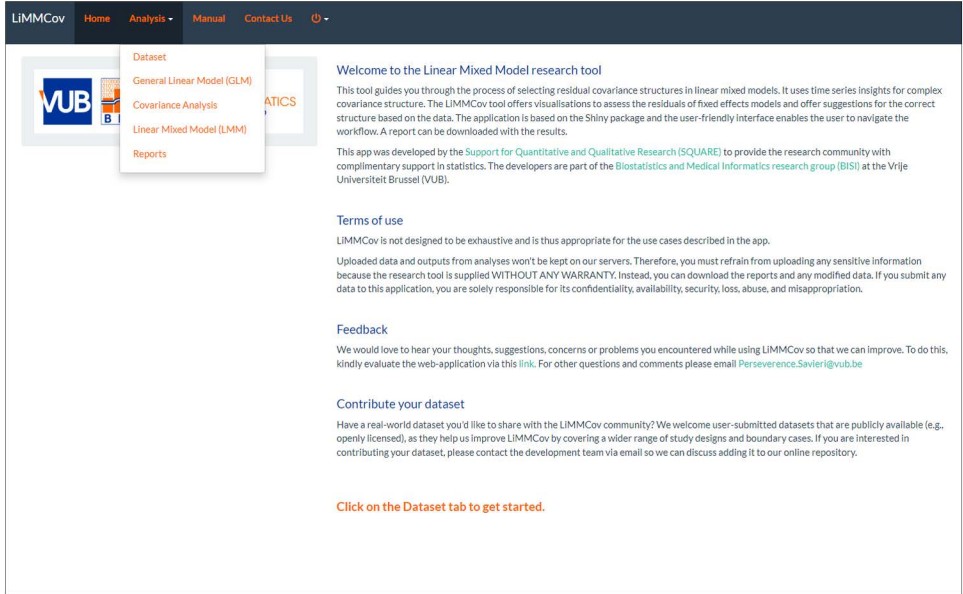

**Fig 3. LiMMCov user interface.** The *Home* page.

capturing the influence of both lag terms in the model. For an AR(p) process, the PACF will have significant spikes up to lag p, after which the values drop to zero, highlighting the true order of the autoregressive model [33].

The *Linear Mixed Model Tab* allows users to specify and fit LMMs to their data. Users can define fixed effects, including interaction terms, and choose from various correlation structures to accurately capture the dependencies in their data. The app uses the **nlme** package [43] to fit these models, which is particularly advantageous as it provides extensive options for modelling the residuals' covariance structures. It allows for the flexible specification of these structures, enabling more precise modelling of the residuals in the data. Thus, LiMMCov offers various covariance structures for the residuals to accommodate different types of data:

- AR1 and AR(p) are designed for discrete and equally spaced time points, as their correlation structures rely on the spacing between observations. Compound Symmetry (CS), while often applied to equally spaced time points, assumes a constant correlation between all observations regardless of time or spacing. This makes CS less sensitive to time spacing and suitable for some discrete, unequally spaced data if the correlation structure remains constant.

- Continuous AR1, Exponential, Linear and Gaussian are for continuous time and are well-suited to irregularly spaced or unbalanced data.

LiMMCov also offers an option for heteroscedastic residual variances via the nlme package's varIdent() function. In the *Linear Mixed Model tab*, users can check the *"Include heteroscedastic variances"* box after selecting a correlation structure (e.g., AR(1)). Once checked, a warning alerts the user that the fitting process may take longer. If the heteroscedastic option is selected, LiMMCov will automatically fit an additional model using the same correlation structure but allowing the residual variance to vary, then present a side-by-side comparison of both the homoscedastic and heteroscedastic fits in the results table. This dynamic workflow ensures that the heteroscedastic model is only estimated when explicitly requested, thereby streamlining the process for users who do not require heteroscedastic variance structures. By including this feature, LiMMCov extends its applicability to real-world longitudinal data where variance may change over time or

across measurement occasions. Additionally, summaries of the results, including model fit comparisons, are provided to confirm the true covariance structure and aid in making informed decisions based on the characteristics of their dataset.

The *Reports* Tab lets users generate and download reports of the output summarising the analyses. Available formats include HTML, PDF, and Word, with HTML recommended for its interactive features. The *Manual Tab* offers a detailed user manual guiding users through each step of using the app, from uploading data, model fitting, and covariance structure selection to report generation. This section helps users understand the functionality of each tab and how to interpret the results. The *Contact Us* Tab provides the contact information of the developers.

## 3. Results

### 3.1 Example: Longitudinal analysis of oxygen saturation in COVID-19 ICU patients

The objective in our example was to model changes in oxygen saturation over time using an LMM and selecting the correct residual covariance structure.

**3.1.1 Load data.** The COVID example dataset was selected from the *Dataset tab,* under the "Choose example data" heading as shown in Fig 4.

**3.1.2 Fit the linear model.** The objective of this initial model was to explore the relationship between oxygen saturation and the fixed effects variables, such as time, age, gender, diastolic BP, and systolic BP. These variables were selected from the *General Linear Model (GLM) tab,* which is part of the *Analysis* menu. On the dropdown menus, users can specify the subject/patient ID, outcome, time and fixed effects before fitting the fixed effects model via the "Run GLM" button (Fig 5).

**3.1.3 Residuals diagnostics.** Residuals were extracted from the fixed effects linear model to analyse the underlying correlation structure. Under the *Covariance Analysis tab*, two key plots were generated (as described in the Methods section):

- *(1) Residual plot* (Fig 6a): The plot shows a pattern of the residuals' downward trend, indicating autoregression.

- *(2) Partial Autocorrelation Function (PACF) plot:* It provides a more precise picture by showing the direct correlation of the residuals with their lagged values, excluding the indirect correlations (Fig 6b). The first three significant lags indicate an AR(3) correlation structure.

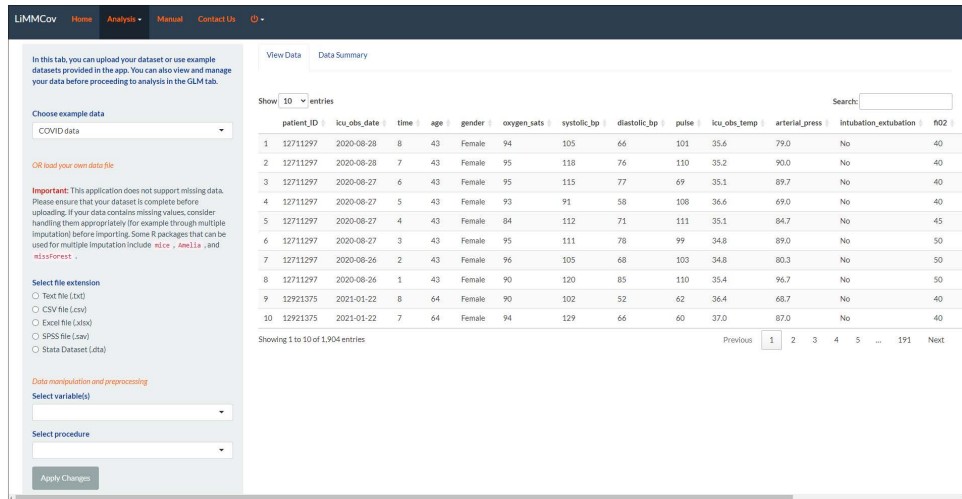

**Fig 4. Dataset tab.** The loaded dataset is displayed with options to preprocess before running the analysis.

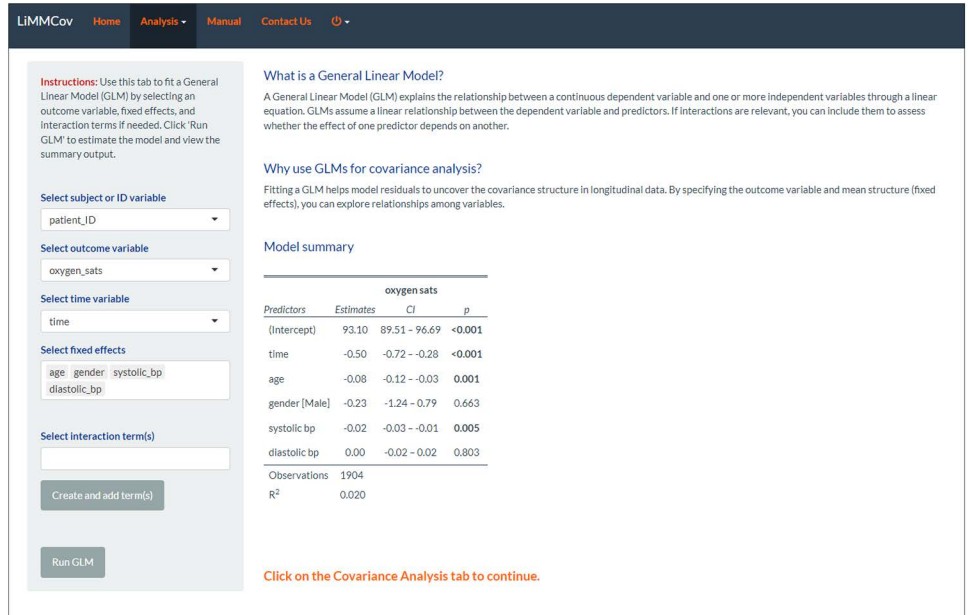

**Fig 5. GLM tab.** The tab shows the fitted linear model and a summary of the coefficients.

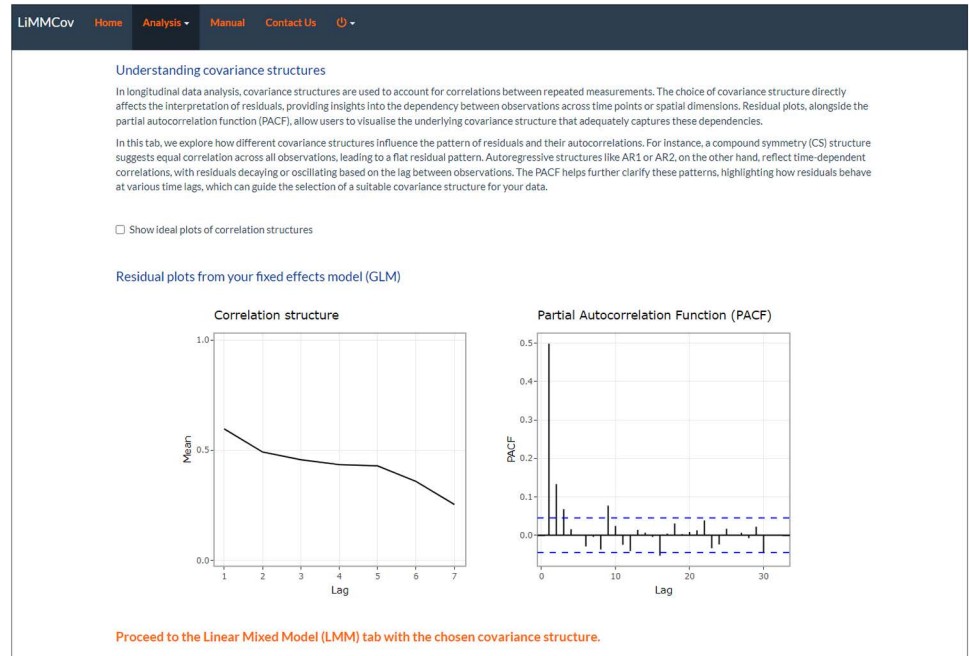

**Fig 6. Covariance Analysis tab.** (a) Residual and (b) PACF plots from the linear model to explore the underlying covariance structure.

**3.1.4 Fit Linear Mixed Model (LMM).** Once the appropriate covariance structure is selected, the linear mixed model can be fitted. Under the *Linear Mixed Model (LMM) tab* (Fig 7), we fitted the model by including the variables from the GLM from Step 2 and selecting the chosen AR(3) covariance structure under the "Select covariance structure" dropdown.

**3.1.5 Final selected model.** Residual plots and the partial autocorrelation function (PACF) in Step 3 suggested an AR(3) covariance structure as the most appropriate for the data. To confirm this, we compared the AR(3) model with alternative covariance structures using AIC, BIC and AICc (Table 1). The AR(3) model had the lowest BIC (13924.97), indicating the best overall fit among the models evaluated. While the AIC values for some models, such as the Symmetric structure, were close, the BIC strongly favoured AR(3), aligning with the visual diagnostics. This result supports the choice of AR(3) as the optimal structure for capturing the correlation patterns in the data.

**3.1.6 Generate Report.** Finally, the app's comprehensive report functionality can be utilised to generate and save the analysis under the *Reports tab* described in section 2.4.

## 4. Discussion

In this study, we present LiMMCov, an interactive tool designed to assist researchers in selecting covariance structures for longitudinal data analyses using linear mixed models. The tool's process includes fitting models with fixed effects, extracting the residuals, and analysing them through residual plots and Partial Autocorrelation Function (PACF) plots. LiMMCov simplifies the user's understanding of residual patterns through interactive visualisations and a user-friendly interface. The tool includes detailed explanations of the common sample covariance structures to guide and support beginner and intermediate users. By leveraging time-series concepts, particularly autoregressive models, LiMMCov effectively identifies complex serial correlation structures. The app allows users to download plot outputs and reports for further analysis or sharing to promote transparency and reproducibility. Using a previously published longitudinal study, we highlighted the app's key features and robust modelling capabilities [42]. Our findings indicate that LiMMCov facilitates the identification of the most appropriate marginal covariance structure, leading to improved model fit and inference. Including interactive visualisations proved crucial in uncovering patterns in the residuals, thereby effectively guiding the selection process. We

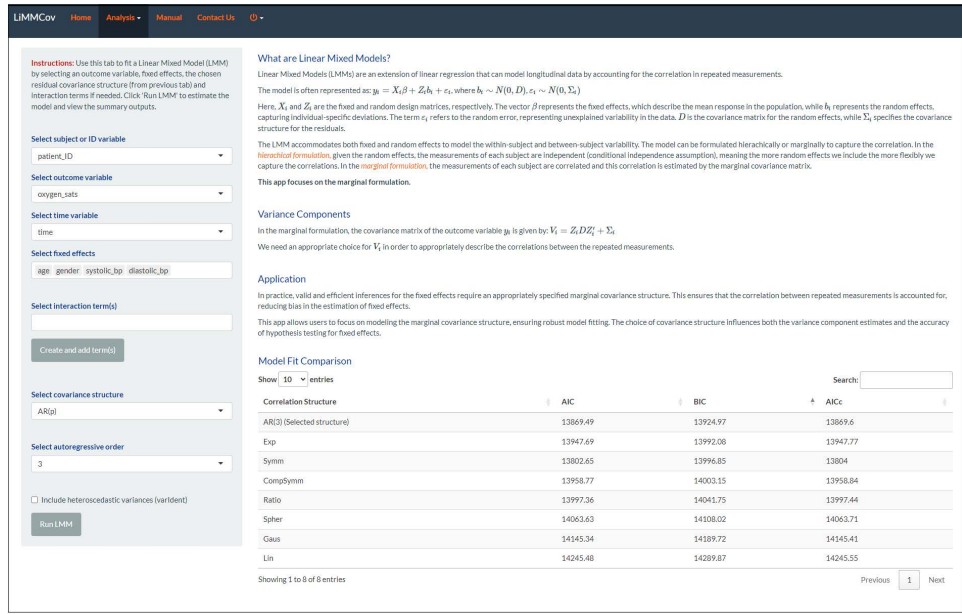

**Fig 7. LMM tab.** The tab shows the fitted model on the sidebar (the summary has been omitted), and the information criteria highlight the best model.

**Table 1. Model fit comparison.**

| Covariance Structure | AIC | BIC | AICc |
|---|---|---|---|
| AR(3) (selected correlation structure) | 13869.49 | 13924.97 | 13869.60 |
| Exponential | 13947.69 | 13992.08 | 13947.77 |
| Symmetry | 13802.65 | 13996.85 | 13804.00 |
| Compound Symmetry | 13958.77 | 14003.15 | 13958.84 |
| Rational quadratics | 13997.36 | 14041.75 | 13997.44 |
| Spherical spatial | 14063.63 | 14108.02 | 14063.71 |
| Gaussian spatial | 14145.34 | 14189.72 | 14145.41 |
| Linear spatial | 14245.48 | 14289.87 | 14245.55 |

validated our results by comparing the model with the selected covariance structure to those fitted with alternative structures available in the **nlme** package using AIC and BIC. In addition, LiMMCov also provides AICc for smaller samples, as recommended in the literature for model selection [12]. Cross-validation methods, while valuable for predictive modelling, fall outside the scope of LiMMCov's primary purpose of identifying appropriate covariance structures. Researchers who need robust variance estimates or wish to evaluate predictive performance may complement LiMMCov analyses with additional external methods. As an open-source application, all code is available on GitHub.

When LiMMCov is compared to existing tools and methodologies, it becomes evident that our approach offers unique advantages, particularly in its user-centric design and focus on interactive visualisation. As Molenberghs and Verbeke [2] highlighted, there is a lack of straightforward methods for comparing various residual covariance structures. Their recommendation to rely on information criteria for model comparison underscores the complexity and manual effort involved in this process and can sometimes obscure the underlying patterns in the data. LiMMCov directly addresses this challenge by offering a platform that automates and enhances these comparisons with interactive visualisations. Although widely used statistical software such as SAS and SPSS can handle linear mixed models for correlated data, these solutions do not offer guidance and are neither free nor intuitive for less experienced users. They also require a solid understanding of model specifications and lack interactive features. Furthermore, LiMMCov focuses specifically on temporal correlations in longitudinal data rather than spatial correlations, which often require different analytical approaches. Spatial correlation methods, such as semi-variograms, have been extended to model random effects [2,44,45]. However, these are outside the scope of LiMMCov, which offers a specialized method that complements (rather than duplicates) spatial analysis techniques. Other Shiny applications, such as rmcorrShiny and SISSREM [31,46], are tools for specific statistical tasks like repeated measures ANOVA. They do not provide an option for selecting the covariance structure, which is a crucial feature of LiMMCov. This distinction highlights LiMMCov's unique capability in addressing the complexities of longitudinal data analysis through visual diagnostic tools, allowing users to observe residual patterns and make informed decisions based on interactive plots. This emphasis on visual and dynamic exploration sets LiMMCov apart from other tools, providing a more accessible and intuitive way for users to engage with complex modelling processes. Other tools often require manual coding or are less suited for exploratory analysis, which LiMMCov addresses effectively by offering a streamlined, no-code interface.

Despite its strengths, we should acknowledge certain limitations of LiMMCov. While the research tool simplifies the selection of the residual covariance structure, it assumes that users have already identified an appropriate mean structure for their linear models. The accuracy of the final model depends not only on the correct specification of the covariance structure but also on the mean structure [2]. Although the time-series concepts applied in LiMMCov require complete data, we recommend that users handle missing data before importing their dataset into the application. For instance, multiple imputation (MI) or other robust techniques can be used to address missing values, ensuring the temporal structure is preserved when necessary [47–50]. In addition, we have added a note and references to the relevant literature [47–50]

in the 'Dataset' and 'Manual' tabs of the LiMMCov app to remind users to address missing data before using the app. Although LiMMCov has demonstrated reliable performance in datasets with at least three repeated measures per subject and a sample size of 50 participants or more, its use in studies with fewer time points (e.g., two or three) remains less thoroughly tested. In these smaller designs, time-series diagnostics (particularly the PACF) may lack stability, and compound symmetry (CS) may be the most practical correlation structure. Consequently, further validation is recommended for datasets with very few repeated measures to ensure robust model selection. We also acknowledge that more complex correlation patterns, such as moving-average (MA) or ARMA(p,q) processes, are currently not available in LiMMCov, as the app focuses on commonly used covariance structures (e.g., AR(p), CS). Although we have begun addressing heteroscedasticity by integrating the varIdent() function for certain covariance structures in LiMMCov, more advanced or generalised variance function classes remain potential avenues for future development. In addition, future work could explore conditional heteroscedastic models, such as ARCH or GARCH, to accommodate volatility clustering, allowing LiMMCov to handle cases where residual variance evolves over time. Evaluating how best to integrate these models into the LMM framework is a promising direction for further research.

Our validation study relied on the Yule-Walker equations to obtain autoregressive coefficients from stationary AR(p) processes; these coefficients then defined the correlation matrices used to simulate longitudinal data. One limitation which deserves mention is that, since Yule-Walker relies on the sample autocorrelation function, parameter estimates can be imprecise in small samples. Although our simulations employed large series to minimise this error, recognising these constraints ensures that the simulated scenarios used to benchmark LiMMCov remain realistic while highlighting considerations for future work that uses empirically derived covariance structures.

A further open issue involves stochastic (unit-root) trends, where the series follows an integrated process (e.g., random walk) rather than the weak-stationary behaviour assumed by the covariance structures currently implemented in LiMMCov. If such a trend is present, researchers should first model it by including time-related fixed effects, so that the residuals analysed by LiMMCov are approximately stationary. At present, the app does not model non-stationary residual behaviour (i.e., the "I" component of an ARIMA(p,d,q) process with $d > 0$), nevertheless, incorporating state-space or other methods that treat integrated components directly is a promising avenue for future development, especially for data in which long-term stochastic trends cannot be captured adequately in the mean structure alone. Evaluating their added value in practice will first require a dedicated feasibility or sensitivity study within the current LMM framework, which we identify as future work.

Finally, specifying both a sophisticated random-effects structure and a serial correlation term in the residual covariance matrix can pose computational challenges and risk non-convergence, particularly with smaller or more complex datasets. Random effects inherently capture some degree of correlation among repeated measurements, so including extensive random effects while also modelling detailed serial correlations may lead to overparameterization or local optima. Consequently, model selection in linear mixed models remains a process of balancing interpretability, convergence, and fit. Although LiMMCov utilises empirical criteria (e.g., AIC, BIC) combined with diagnostic plots to aid model building, it does not guarantee the "true" or most complex model; rather, it helps researchers arrive at a reasonable and justifiable specification grounded in the observed data. Notably, LiMMCov follows a marginal formulation perspective rather than a broader hierarchical model-building approach. This aligns with standard model-building principles, wherein the final choice often reflects both data-driven insights and practical constraints, such as sample size and computational feasibility.

## 5. Conclusion

In summary, the current manuscript presents a solution to the challenges researchers face when using linear mixed models in longitudinal data analysis. LiMMCov represents a significant advancement in the tools available for LMM analysis for selecting covariance structures. Providing an interactive, user-friendly platform empowers researchers to make more informed decisions, ultimately improving the quality and reliability of their analyses.

## Supporting information

**S1 File. Appendix A.** Yule-Walker equations for AR(1) and AR(2) processes.
(DOCX)

**S2 File. Appendix B.** R Code for longitudinal data simulation and estimation of AR parameters.
(DOCX)

## Acknowledgments

We thank Peter S. Nyasulu from the Division of Epidemiology and Biostatistics, Faculty of Medicine and Health Sciences, Stellenbosch University (Cape Town, South Africa), for providing the example dataset.

## Author contributions

**Conceptualization:** Perseverence Savieri, Lara Stas, Kurt Barbé.

**Formal analysis:** Perseverence Savieri.

**Methodology:** Perseverence Savieri, Lara Stas, Kurt Barbé.

**Software:** Perseverence Savieri.

**Supervision:** Lara Stas, Kurt Barbé.

**Validation:** Perseverence Savieri.

**Visualization:** Perseverence Savieri.

**Writing – original draft:** Perseverence Savieri.

**Writing – review & editing:** Perseverence Savieri, Lara Stas, Kurt Barbé.

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
