## [Decision Letter · Decision Letter 0]

2 Feb 2019

Dear Dr. Savieri,

Thank you for submitting your manuscript to PLOS ONE. After careful consideration, we feel that it has merit but does not fully meet PLOS ONE’s publication criteria as it currently stands. Therefore, we invite you to submit a revised version of the manuscript that addresses the points raised during the review process.

We look forward to receiving your revised manuscript.

Kind regards,

Mohamed R. Abonazel, Ph.D.

Academic Editor

PLOS ONE

Journal Requirements:

2. We note that Figures 3,4,5,6 and 7 in your submission contain copyrighted images. All PLOS content is published under the Creative Commons Attribution License (CC BY 4.0), which means that the manuscript, images, and Supporting Information files will be freely available online, and any third party is permitted to access, download, copy, distribute, and use these materials in any way, even commercially, with proper attribution. For more information, see our copyright guidelines: http://journals.plos.org/plosone/s/licenses-and-copyright.

1. You may seek permission from the original copyright holder of Figures 3,4,5,6 and 7 to publish the content specifically under the CC BY 4.0 license.

Reviewers' comments:

Reviewer's Responses to Questions

**Comments to the Author**

1. Is the manuscript technically sound, and do the data support the conclusions?

Reviewer #1: Yes

2. Has the statistical analysis been performed appropriately and rigorously?

Reviewer #1: Yes

3. Have the authors made all data underlying the findings in their manuscript fully available?

Reviewer #1: Yes

4. Is the manuscript presented in an intelligible fashion and written in standard English?

Reviewer #1: Yes

Reviewer #1: Peer Review of “LiMMCov: An Interactive Research Tool for Efficiently Selecting Covariance Structures in Linear Mixed Models Using Insights from Time Series Analysis”

I would like to congratulate the authors on this important topic. The manuscript addresses a real and persistent challenge in longitudinal data analysis: correctly specifying the residual covariance structure in linear mixed models (LMMs). By offering an interactive application (LiMMCov) that leverages both standard information criteria (AIC, BIC, etc.) and time‐series diagnostics (residual plots, PACF), the paper tackles a topic of clear importance for applied researchers in fields such as medicine, epidemiology, and social sciences.

Incorporating time‐series perspectives (e.g., Yule‐Walker equations, partial autocorrelation function plots) into the LMM covariance‐structure selection process is a distinct and innovative feature. Despite their proven effectiveness in detecting autoregressive patterns, these concepts are often underutilised in longitudinal analyses.

The manuscript clearly describes how LiMMCov can be used point-and-click, requires minimal coding, and provides direct visual feedback (residual plots, PACF plots, etc.). This format lowers the barrier to advanced modelling for practitioners and graduate students who may not otherwise implement thorough covariance-structure checks.

The code's availability on GitHub and free deployment in R Shiny are marked advantages. They foster transparency, reproducibility, and collaboration—principles strongly encouraged in scientific research.

Major areas of improvement

1- Real‐world longitudinal studies often have missing time points or irregular sampling intervals. While the manuscript mentions that LiMMCov “requires complete data,” it would benefit from guidance or references on handling missingness before importing.

2- Although the authors note that performance was not “thoroughly tested” for data sets with few repeated measures or small samples, this point warrants stronger emphasis. In practice, many longitudinal studies have 2–3 time points. The utility and stability of time‐series diagnostics (particularly PACF) in such constrained designs might be limited.

3- While combining AIC/BIC with diagnostic plots is valuable, the paper might clarify the limitations of purely empirical selection (e.g., the possibility of local optima or how random effects can alter apparent residual patterns). Specifically, clarifying how random‐effects structures and residual covariance structures interact would strengthen the discussion.

4- The manuscript focuses on common information criteria (AIC, BIC) and the partial autocorrelation function. However, other approaches (e.g., cross‐validation, robust standard errors, or specialised small‐sample corrections like AICc) are only briefly mentioned. It might be worth highlighting or demonstrating how these can be integrated within the app or used in tandem with its outputs.

5- Many longitudinal data sets may exhibit more complex correlation patterns (e.g., moving‐average components or partially nested random effects). It would be prudent to acknowledge that LiMMCov currently does not handle MA(q) or ARMA(p,q) structures.

6- As the authors note, random effects and residual covariance structures can be complementary. An overly complicated random‐effects structure could obscure the patterns in the residuals. Readers might need guidance on balancing the complexity of each part of the model.

7- Provide a quick “decision tree” in the manual: for instance, “If your residual plot shows a gradually decaying correlation with lag, consider AR(1). If it shows oscillation, consider AR(2). If a flat correlation for all lags, consider CS,” etc. This can be an excellent at-a-glance guide.

8- Enhance reporting of model diagnostics. In addition to the final AIC/BIC, incorporate:

o Model Convergence Warnings: Indicate if the model has trouble converging, prompting the user to revise initial guesses.

o Random-effects Diagnostics: Although residual correlation is the focus, briefly mentioning random-effects variance components can reassure advanced users about model fit.

9- While the authors discuss covariance structures, many practical problems also feature heteroskedastic residuals (where variance changes over time). Adding a note on how LiMMCov might be extended or used when residual variances are not assumed constant (e.g., varIdent in nlme) would expand its scope.

10- To identify boundary cases, encourage broader usage and external testing (e.g., including more diverse example data sets or user-submitted open data).

Minor issues

1- The word “models” is repeated (“LMMs models”), the word models is in LMMs. Please remove models after LMMs.

2- Lines 84–85 (“However, the AIC does not offer insight ... black box identification”): The sentence could be clarified.

3- While the text defines many terms (e.g., AR(p), CS, etc.), some readers may benefit from succinct definitions or references to standard texts the first time each structure is named. For example, a short descriptive bullet might clarify that “Compound Symmetry (CS) implies a constant correlation ρ between any two observations for the same subject.”

4- The results section might be clearer with subheadings like “3.1 Residual Diagnostics,” “3.2 Model Fit Comparison,” “3.3 Final Selected Model,” to provide an at-a-glance roadmap.

5- Certain sentences would read more fluently if combined or slightly rephrased, especially around lines 468–476, where the text shifts from discussing SAS/SPSS limitations to referencing time-series approaches. Ensure each paragraph transitions smoothly from one concept to the next.

6- Overall, the text is quite readable but can be made more concise by removing repeated points and ensuring that each paragraph targets a single theme.

**Do you want your identity to be public for this peer review?** For information about this choice, including consent withdrawal, please see our Privacy Policy

Reviewer #1: **Yes: ** Osama Abdelhay

---

## [Author Response · Author response to Decision Letter 1]

28 Mar 2025

Reviewer #1:

Major areas of improvement

1- Real‐world longitudinal studies often have missing time points or irregular sampling intervals. While the manuscript mentions that LiMMCov “requires complete data,” it would benefit from guidance or references on handling missingness before importing.

Response: Thank you for this suggestion. We first want to point out that multiple imputation is only one of various strategies for handling missing data, and each approach (e.g., full information maximum likelihood, inverse probability weighting) has specific requirements and assumptions, particularly regarding sample size or missing not-at-random (MNAR) data. We have revised the Discussion section of the manuscript (lines 526-532) to provide references that discuss different missing-data methods in more detail, helping readers choose techniques best suited to their study’s characteristics. In addition, we emphasize that LiMMCov focuses on the selection of covariance structures, so data preprocessing and the choice of imputation or missing-data handling approach lie outside the app’s scope. Nevertheless, we have included notes in the ‘Dataset’ and ‘Manual’ tabs and the app’s manual to remind users to address missingness appropriately (for instance, via multiple imputation or other suitable methods) before uploading data to LiMMCov. This ensures the validity of subsequent analyses and aligns with best practices in longitudinal data handling.

• Molenberghs G, Fitzmaurice G, Kenward MG, Tsiatis A, Verbeke G. Handbook of missing data methodology. CRC Press; 2014.

• Nooraee N, Molenberghs G, Ormel J, Van den Heuvel ER. Strategies for handling missing data in longitudinal studies with questionnaires. J Stat Comput Simul [Internet]. 2018;88(17):3415–36. Available from: https://doi.org/10.1080/00949655.2018.1520854

• Ji L, Chow SM, Schermerhorn AC, Jacobson NC, Cummings EM. Handling Missing Data in the Modeling of Intensive Longitudinal Data. Struct Equ Model [Internet]. 2018;25(5):715–36. Available from: https://doi.org/10.1080/10705511.2017.1417046

• Heymans MW, Twisk JWR. Handling missing data in clinical research. J Clin Epidemiol [Internet]. 2022;151:185–8. Available from: https://doi.org/10.1016/j.jclinepi.2022.08.01

2- Although the authors note that performance was not “thoroughly tested” for data sets with few repeated measures or small samples, this point warrants stronger emphasis. In practice, many longitudinal studies have 2–3 time points. The utility and stability of time‐series diagnostics (particularly PACF) in such constrained designs might be limited.

Response: We appreciate the reviewer’s focus on studies with minimal time points. We have revised the manuscript (lines 533-543) to emphasize more strongly that the stability and interpretability of time-series diagnostics (such as the PACF) are indeed limited when the number of repeated measures is very low. In these scenarios, a compound symmetry structure may be the most practical option. Our simulations suggest that LiMMCov performs well with at least three repeated measures per subject and a sample size of 50 participants or more, but we acknowledge the need for further research to confirm performance in smaller designs. We believe these clarifications better highlight the app’s limitations in real-world settings with constrained data.

3- While combining AIC/BIC with diagnostic plots is valuable, the paper might clarify the limitations of purely empirical selection (e.g., the possibility of local optima or how random effects can alter apparent residual patterns). Specifically, clarifying how random‐effects structures and residual covariance structures interact would strengthen the discussion.

Response: We appreciate your attention to this important interplay between random effects and residual covariance structures. We have added a paragraph (lines 550-564) in the Discussion section acknowledging that specifying multiple or complex random effects can already capture part of the serial correlation, making the simultaneous specification of a detailed marginal covariance structure more prone to overparameterization or non-convergence. Moreover, we emphasize that our approach offers a practical, data-driven mechanism to arrive at a defendable model, though not necessarily the most complex or “true” model. This recognizes the inherent limitations of empirical selection and underscores that LiMMCov should be used as part of a careful model-building process that balances fit, interpretability, and feasibility in real-world datasets.

4- The manuscript focuses on common information criteria (AIC, BIC) and the partial autocorrelation function. However, other approaches (e.g., cross‐validation, robust standard errors, or specialised small‐sample corrections like AICc) are only briefly mentioned. It might be worth highlighting or demonstrating how these can be integrated within the app or used in tandem with its outputs.

Response: We acknowledge the reviewer’s interest in additional model selection and diagnostic approaches. In response, we now include AICc in LiMMCov’s model-fit comparison table, specifically for smaller-sample scenarios where AICc may be more appropriate than AIC or BIC. Regarding cross-validation, we believe it is more aligned with predictive objectives, whereas LiMMCov’s primary goal is to identify and compare different covariance structures in a longitudinal framework. As such, we currently do not implement cross-validation. We acknowledge that robust standard errors could provide further insights when model assumptions are questioned, but this extends beyond the scope of our tool, which emphasizes comparative model diagnostics rather than fully robust estimation. Nonetheless, we have expanded the Discussion section (lines 484-490) to note that users may apply robust standard errors or cross-validation in external workflows, potentially in conjunction with LiMMCov output, if their study objectives require these additional methods.

5- Many longitudinal data sets may exhibit more complex correlation patterns (e.g., moving‐average components or partially nested random effects). It would be prudent to acknowledge that LiMMCov currently does not handle MA(q) or ARMA(p,q) structures.

Response: Thank you for highlighting this. We have added a note in the Discussion (lines 540-543) explicitly acknowledging that LiMMCov does not currently support MA(q) or ARMA(p,q) structures. We aimed to focus on the most commonly employed covariance formulations (e.g., AR(p), CS) to keep the interface and model selection process manageable for most typical longitudinal studies. Nonetheless, we acknowledge the need for more advanced correlation structures in future updates.

6- As the authors note, random effects and residual covariance structures can be complementary. An overly complicated random‐effects structure could obscure the patterns in the residuals. Readers might need guidance on balancing the complexity of each part of the model.

Response: We appreciate the reviewer’s emphasis on finding the right balance between random effects complexity and the modelling of residual correlations. However, LiMMCov follows the marginal formulation perspective, guiding the marginal covariance structure selection. We acknowledge this limitation by linking it to comment (3). In the revised Discussion (lines 550-564), we clarify that LiMMCov is designed specifically to help users select among candidate residual covariance structures and does not handle the full hierarchical model-building process, which includes random effects. We note that excessive complexity in the random-effects portion can mask meaningful residual correlations..

7- Provide a quick “decision tree” in the manual: for instance, “If your residual plot shows a gradually decaying correlation with lag, consider AR(1). If it shows oscillation, consider AR(2). If a flat correlation for all lags, consider CS,” etc. This can be an excellent at-a-glance guide.

Response: We appreciate this suggestion to offer a concise “decision tree” for guiding users. We have incorporated such a guide into the “Manual” tab of the LiMMCov application, where users can find recommendations on when to consider AR(1), AR(2), Compound Symmetry, or other structures based on their residual plots and autocorrelation patterns. We believe this resource enhances the user experience by clarifying how to interpret diagnostic outputs and choose an appropriate correlation structure more confidently.

8- Enhance reporting of model diagnostics. In addition to the final AIC/BIC, incorporate:

o Model Convergence Warnings: Indicate if the model has trouble converging, prompting the user to revise initial guesses.

o Random-effects Diagnostics: Although residual correlation is the focus, briefly mentioning random-effects variance components can reassure advanced users about model fit.

Response: We appreciate the reviewer’s suggestions and are pleased to confirm that LiMMCov already provides checks and warnings when models fail to converge, prompting users to adjust their inputs accordingly. Regarding random-effects diagnostics, we respectfully note that LiMMCov is rooted in the marginal formulation and thus focuses on modelling the residual covariance matrix rather than examining different random-effects structures in detail (as also discussed in our responses to Comments 3 and 6). In the Discussion section, we acknowledge that random-effects variance component diagnostics can be valuable for advanced users, but these features currently lie outside the scope of our tool’s primary objective.

9- While the authors discuss covariance structures, many practical problems also feature heteroskedastic residuals (where variance changes over time). Adding a note on how LiMMCov might be extended or used when residual variances are not assumed constant (e.g., varIdent in nlme) would expand its scope.

Response: We appreciate the reviewer’s observation regarding heteroskedastic residuals. Although our approach initially assumes stationarity, we have added an option in LiMMCov to allow for both homoscedastic and heteroscedastic (varIdent()) residual variances for the selected covariance structure. This update acknowledges that real-world longitudinal data often exhibits changing variance over time. However, extending LiMMCov to a broader range of heteroscedastic models offered under variance function classes (i.e. varClasses in nlme) is currently out of scope (lines 543-548).

10- To identify boundary cases, encourage broader usage and external testing (e.g., including more diverse example data sets or user-submitted open data).

Response: Thank you for this suggestion. In the current version, we provide multiple example datasets, both simulated (AR1, AR2, CS) and from a published study, to demonstrate typical use cases. We also allow users to upload their own datasets with varying designs and complexities. To further expand testing and identify boundary cases not captured by our initial examples, we welcome user-submitted open data, and we have added a not in the app to encourage those willing to make their datasets publicly available to share them. We plan to maintain an online repository of such user-contributed datasets, enabling broader community engagement and facilitating more extensive real-world testing of LiMMCov.

Minor issues

1- The word “models” is repeated (“LMMs models”), the word models is in LMMs. Please remove models after LMMs.

Response: Thank you for noticing this; we have removed the word models after LMMs (lines 1-2).

2- Lines 84–85 (“However, the AIC does not offer insight ... black box identification”): The sentence could be clarified.

Response: Thank you for pointing out the need for clarification. We have revised the sentence to emphasize that although AIC provides a numerical basis for comparing models, it does not reveal the underlying reasons why one model’s covariance structure may be preferable. This lack of interpretability can be problematic for researchers who wish to understand the justification behind the chosen model. The revised text now reads (lines 87-89): “However, while AIC is useful for comparing competing models, it offers limited insight into why one model outperforms another, effectively making the selection process a ‘black box’ for identifying the model structure.”

3- While the text defines many terms (e.g., AR(p), CS, etc.), some readers may benefit from succinct definitions or references to standard texts the first time each structure is named. For example, a short descriptive bullet might clarify that “Compound Symmetry (CS) implies a constant correlation ρ between any two observations for the same subject.”

Response: Thank you for highlighting that more detail or references would be helpful when introducing each covariance structure. We have revised the manuscript by adding a concise definition and relevant references the first time each structure is mentioned (lines 55-60). This should assist readers who are less familiar with AR(1), compound symmetry (CS), and other covariance structures in understanding their basic assumptions and where to find further details.

4- The results section might be clearer with subheadings like “3.1 Residual Diagnostics,” “3.2 Model Fit Comparison,” “3.3 Final Selected Model,” to provide an at-a-glance roadmap.

Response: Thank you for suggesting clearer subheadings in the Results section. We have reorganized the presentation to include subheadings such as “Residual Diagnostics,” and “Final Selected Model,” while retaining the logical steps to illustrate how the Shiny app is used from data loading to final report generation. This approach should provide readers with both a step-by-step workflow and an at-a-glance roadmap of the key analytical steps, as recommended.

5- Certain sentences would read more fluently if combined or slightly rephrased, especially around lines 468–476, where the text shifts from discussing SAS/SPSS limitations to referencing time-series approaches. Ensure each paragraph transitions smoothly from one concept to the next.

Response: Thank you for your suggestion. We have revised the manuscript, and the sentences around these lines (now lines 500-509) have been rephrased to improve their fluency and transitions.

6- Overall, the text is quite readable but can be made more concise by removing repeated points and ensuring that each paragraph targets a single theme.

Response: Thank you. We have revised the manuscript to address repeated points and ensure that each paragraph targets a single theme. Specifically, we consolidated overlapping discussions, removed redundant sentences, and reorganized the text to improve clarity and flow, as can be seen in the track changes. We believe these changes have resulted in a more concise and coherent manuscript.

---

## [Decision Letter · Decision Letter 1]

29 Apr 2025

Dear Dr. Savieri,

Thank you for submitting your manuscript to PLOS ONE. After careful consideration, we feel that it has merit but does not fully meet PLOS ONE’s publication criteria as it currently stands. Therefore, we invite you to submit a revised version of the manuscript that addresses the points raised during the review process.

We look forward to receiving your revised manuscript.

Kind regards,

Mohamed R. Abonazel, Ph.D.

Academic Editor

PLOS ONE

Journal Requirements:

Reviewers' comments:

Reviewer's Responses to Questions

**Comments to the Author**

Reviewer #1: All comments have been addressed

Reviewer #2: All comments have been addressed

2. Is the manuscript technically sound, and do the data support the conclusions?

Reviewer #1: Yes

Reviewer #2: Yes

3. Has the statistical analysis been performed appropriately and rigorously?

Reviewer #1: Yes

Reviewer #2: Yes

4. Have the authors made all data underlying the findings in their manuscript fully available?

Reviewer #1: Yes

Reviewer #2: Yes

5. Is the manuscript presented in an intelligible fashion and written in standard English?

Reviewer #1: Yes

Reviewer #2: Yes

Reviewer #1: Dear authors, congratulations on this important manuscript. Your developed tool is very useful and have practical implementation to any researcher working with longitudinal studies.

Reviewer #2: The article seems interesting and proposes software solutions for the specific problems of Mixed Models, particularly for longitudinal analysis over time.

However, I do have some reservations regarding the material presented:

1. The Yule-Walker equations have some limitations that the authors should consider:

The accuracy of the estimates depends on the assumption that the time series is stationary, meaning that its statistical properties do not change over time. If the time series is non-stationary, differencing or other transformations may be required before applying the Yule-Walker method.

The Yule-Walker estimates are based on the sample autocorrelation function, which can introduce sampling error, especially for small sample sizes.

2. The PACF function in Figure 6 has several values that exceed the confidence interval limits at lags 9 and 16. The component's behavior decreases harmonically, which suggests — based on the inversion of AR models into MA models — the possibility of the existence of an MA component rather than higher-order AR components.

3. The authors correctly note that there are still "gray areas" to be explored, but I would recommend also discussing the possibility of a stochastic trend.

4. The violation of stationarity in the series variance — which is closely linked to the covariance matrix — should not be overlooked. This leads to the use of ARCH models, which opens a new “window of opportunity” for future articles. This aspect should also be mentioned among future research directions!

I would like to congratulate and encourage the authors for their effort in developing and aggregating existing methods for time series analysis within the framework of mixed models!

**Do you want your identity to be public for this peer review?** For information about this choice, including consent withdrawal, please see our Privacy Policy

Reviewer #1: **Yes: ** Osama Abdelhay

Reviewer #2: No

---

## [Author Response · Author response to Decision Letter 2]

6 May 2025

Manuscript ID: PONE-D-25-03886R1

Manuscript title: LiMMCov: An Interactive Research Tool for Efficiently Selecting Covariance Structures in Linear Mixed Models Using Insights from Time Series Analysis

Reviewer #2:

Minor areas of improvement

1. The Yule-Walker equations have some limitations that the authors should consider:

The accuracy of the estimates depends on the assumption that the time series is stationary, meaning that its statistical properties do not change over time. If the time series is non-stationary, differencing or other transformations may be required before applying the Yule-Walker method.

The Yule-Walker estimates are based on the sample autocorrelation function, which can introduce sampling error, especially for small sample sizes.

Thank you for pointing this out. From a linear mixed-model viewpoint, the residual covariance structures that LiMMCov compares (i.e, AR(1), AR(p), compound symmetry, etc.) from the nlme package (Pinheiro et al., 2023), all embed the assumption of weak stationarity. Because this assumption underlies the likelihood for any LMM fitted with such structures, it was natural to maintain it when generating synthetic data for our validation study. Therefore, we first generated stationary AR(p) processes (verifying that all characteristic roots lay outside the unit circle) and then applied the Yule-Walker equations to obtain the corresponding AR coefficients. These coefficients defined the correlation matrices used to simulate longitudinal datasets for benchmarking LiMMCov. We have clarified in the Methods section (lines 223-225) that the Yule-Walker procedure was applied under weak-stationarity assumptions. We also acknowledge, now in the Discussion (lines 553-561), that Yule-Walker estimators rely on the sample autocorrelation function and can be imprecise in very small samples. However, this potential sampling error pertains only to the data-generation phase of our simulations; LiMMCov itself remains agnostic to how the covariance matrix is obtained and can be applied to empirical data regardless of the method used to estimate or specify residual correlations. Hence, the core functionality of the app is not altered.

2. The PACF function in Figure 6 has several values that exceed the confidence interval limits at lags 9 and 16. The component's behavior decreases harmonically, which suggests — based on the inversion of AR models into MA models — the possibility of the existence of an MA component rather than higher-order AR components.

We appreciate the reviewer’s time-series perspective. The out-of-band spikes at lags 9 and 16 indeed suggest that a moving-average (MA) component, or an ARMA(p,q) process, could underlie the residual pattern. LiMMCov, however, is intentionally limited to autoregressive and other commonly used stationary covariance structures in linear mixed models; modelling explicit MA or ARMA components is beyond this first release's scope. As noted in the Discussion (and echoed by Reviewer 1), the absence of MA/ARMA covariance structures is a current limitation (lines 541-544) and an important direction for future development as we continue integrating time-series methods into LMM frameworks. In the meantime, users who suspect MA behaviour can either analyse the series externally with dedicated time-series software or approximate the effect by increasing the AR order within LiMMCov, while recognising the approximation’s limitations.

3. The authors correctly note that there are still "grey areas" to be explored, but I would recommend also discussing the possibility of a stochastic trend.

We appreciate this insightful recommendation. LiMMCov is intended as an entry point for the linear-mixed-model (LMM) community to adopt time-series concepts; accordingly, the current version assumes that, once an appropriate mean structure is specified, the residuals are weakly stationary. Any stochastic (unit-root) trend, therefore, needs to be removed or modelled in the fixed-effects part of the LMM (e.g., polynomial time terms) before LiMMCov’s covariance-structure selection is applied. Because an ARIMA(p,d,q) process with d>0 becomes an ARMA(p,q) process after differencing, such non-stationary behaviour cannot be accommodated simply by specifying an “ARIMA covariance matrix.” Supporting integrated components would instead require additional modelling layers (e.g., automatic differencing or state-space extensions), and evaluating their usefulness within the LMM framework would first necessitate a feasibility or sensitivity study, work we regard as future research rather than part of the present release. We have revised the Discussion section (lines 541-544 and 563–573) with parts that (i) explain how users can handle a stochastic trend with the existing tool and (ii) identify the incorporation of extended ARMA-type covariance structures as a logical, but still prospective, enhancement to LiMMCov.

4. The violation of stationarity in the series variance — which is closely linked to the covariance matrix — should not be overlooked. This leads to the use of ARCH models, which opens a new “window of opportunity” for future articles. This aspect should also be mentioned among future research directions!

We agree that conditional heteroskedasticity, often observed as “volatility clustering,” where large residuals tend to be followed by large residuals, constitutes an important avenue for future work. At present, LiMMCov can accommodate unconditional heteroskedasticity via varIdent(), but it does not estimate models in which the residual variance evolves as an explicit stochastic process driven by past squared errors (e.g., ARCH/GARCH behaviour). We have therefore added a sentence in the Discussion (lines 547-551) that (i) acknowledges volatility clustering as a motivation for ARCH models and (ii) lists ARCH/GARCH covariance structures as a promising avenue for future development and methodological research.

---

## [Decision Letter · Decision Letter 2]

LiMMCov: An Interactive Research Tool for Efficiently Selecting Covariance Structures in Linear Mixed Models Using Insights from Time Series Analysis

PONE-D-25-03886R2

20 May 2025

Dear Dr. Savieri,

We’re pleased to inform you that your manuscript has been judged scientifically suitable for publication and will be formally accepted for publication once it meets all outstanding technical requirements.

Kind regards,

Mohamed R. Abonazel, Ph.D.

Academic Editor

PLOS ONE

Additional Editor Comments (optional):

Reviewers' comments:

Reviewer's Responses to Questions

**Comments to the Author**

Reviewer #2: All comments have been addressed

2. Is the manuscript technically sound, and do the data support the conclusions?

Reviewer #2: Yes

3. Has the statistical analysis been performed appropriately and rigorously?

Reviewer #2: Yes

4. Have the authors made all data underlying the findings in their manuscript fully available?

Reviewer #2: Yes

5. Is the manuscript presented in an intelligible fashion and written in standard English?

Reviewer #2: Yes

Reviewer #2: (No Response)

**Do you want your identity to be public for this peer review?** For information about this choice, including consent withdrawal, please see our Privacy Policy

Reviewer #2: No

---

## [Editor Report · Acceptance letter]

PONE-D-25-03886R2

PLOS ONE

Dear Dr. Savieri,

I'm pleased to inform you that your manuscript has been deemed suitable for publication in PLOS ONE. Congratulations! Your manuscript is now being handed over to our production team.

Kind regards,

on behalf of

Dr Mohamed R. Abonazel

Academic Editor

PLOS ONE